

# Brief Communication: Rapid acceleration of the Brunt Ice Shelf after calving of iceberg A-81

Oliver J. Marsh[1], Adrian J. Luckman[2], Dominic A. Hodgson[1]

[1]British Antarctic Survey, Cambridge, UK

[2]Department of Geography, Swansea University, Swansea, UK

*Correspondence to*: Oliver J. Marsh (olrs@bas.ac.uk)

**Abstract.** The Brunt Ice Shelf, Antarctica, accelerated rapidly from 900 m a$^{-1}$ to 1500 m a$^{-1}$ during six months following the calving of a 1500 km$^2$ iceberg on 22$^{nd}$ January 2023. Initially, the rate of acceleration increased by a factor of ten, with a second, smaller calving at the end of June 2023 leading to further tripling of acceleration. The acceleration is caused by

reduction of buttressing at the McDonald Ice Rumples due to loss of contact with the sea floor and has led to high strain rates to the south, with potential consequences for the stability of the remaining ice shelf.

## 1 Introduction

The Brunt Ice Shelf is the westernmost of the ice shelves which line the coast of East Antarctica. It has a 60-year record of glaciological observations which show velocities ranging from 400 to 800 m a$^{-1}$ (Gudmundsson, 2017). Historical variability

in velocity is inferred to be primarily controlled by the coupling between the ice shelf and the sea floor at a pinning point known as the McDonald Ice Rumples (MIR) (Thomas, 1971, Hodgson et al., 2019), and to follow a cyclical pattern (De Rydt et al., 2019). Through the late 1990s and early 2000s, a strengthening of the ice-bed contact led to reduction in ice velocity across the ice shelf and increased stress in the vicinity of the MIR. Since 2013, crevassing has increased around the MIR, weakening the ice shelf in this area, and several new rifts have formed or re-activated including Chasm-1, Halloween Crack

and North Rift (Fig. 2a). The growth of these cracks culminated in the recent calving of two large icebergs – A-74 in February 2021 and A-81 in January 2023 (Francis et al., 2022; Morris et al., 2023).

Modelling and observations show that ice shelf thinning around Antarctica drives acceleration (Reese et al., 2018), but there are fewer observations of how tabular calving impacts ice shelves because large calving events are rare. Here we present the

most recent GPS and satellite observations of the Brunt Ice Shelf since the A-81 iceberg calving and discuss the mechanisms responsible for the recent acceleration and the potential consequences for the remaining ice shelf.





## 2 Methods

### 2.1 GPS

GPS velocity data is presented for ten consecutive years (2013-2023) from three sites on the Brunt Ice Shelf within the vicinity
of the current Halley VI Research Station site (Fig. 1). The GPS at site LL20 recorded from 2013-2017. GPS at ZZ6A
recorded from 2017 onwards and was originally located approximately five kilometres east of the final position of LL20 (Fig.
2c). Velocity differences between sites are less than 3% and no adjustment is made to account for advection of the GPS
receivers. Data from both sites were recorded for two hours per day at 30-second intervals using a dual-frequency Leica GS10,
subsampled to 300-second intervals and processed into positions using Natural Resources Canada's Precise Point Positioning
(CSRS-PPP) software. Velocities are then calculated from the gradient of a linear fit through polar stereographic x and y
positions using a running fitting window of 14-days. The GPS at site ZMET was located approximately 500 m west of ZZ6A
and recorded from 2022 onwards for 24-hours per day at one-second interval. Data are subsampled and processed in the same
way as for ZZ6A and LL20, although with a fitting window of one-day to highlight short-term variability.

### 2.2 Feature Tracking

Two-dimensional surface velocities (Fig. 3a, b) were derived by feature-tracking between pairs of Sentinel-1A images
separated by 12-days using standard methods (e.g., Luckman et al., 2015) and the Gamma software package. Velocity maps
were projected to the Polar Stereographic coordinate system (EPSG:3031) at a resolution of 100 m. Image pairs were chosen
to maximise spatial coverage of valid velocity measurements and to span the maximum duration since the calving of A-81.
Magnitudes of first principal strain rates (Fig. 3e, f) were computed on a 3x3 grid.

## 3 Results

### 3.1 Velocity Increase

From 2013 to 2023, ice velocity measured by GPS at the current site of Halley VI Research Station (ZZ6A) gradually increased
from around 450 m a$^{-1}$ to 900 m a$^{-1}$ at a uniform rate of 50 m a$^{-2}$ over ten years (Fig. 1a). During this period, there was a shift
from radial crevassing (resulting from buttressing) to chaotic shear damage around the McDonald Ice Rumples (Fig. 2d-e),
Chasm-1 (Fig. 2a) propagated towards the MIR (De Rydt et al., 2018) and Halloween Crack (Fig. 2a) widened (Morris et al.,
2023). GPS data throughout this period shows a tidal variation in velocity of the ice shelf of around ±10% over a fortnightly
period, synchronous with the Msf tide (with velocity peaks during spring tide). Large, semi-diurnal variations in flow have
previously been observed on the Brunt Ice Shelf (Doake et al., 2002) and the tidally-driven mechanism that results in non-
linear fortnightly behaviour has been observed elsewhere in the Weddell Sea (Gudmundsson, 2006; Rosier and Gudmundsson,
55 2020).



The calving of A-74 in February 2021 did not lead to a substantial change in velocity at ZZ6A, as the North Rift (Fig. 2a) did not interact with the grounded parts of the ice shelf to the south of the MIR. In contrast, there was a distinct change in dynamics following the calving of A-81, with the rate of acceleration increasing by an order of magnitude (Fig. 1). After accounting for the periodic tidal behaviour, there is an apparent immediate shift from a constant rate of acceleration of 50 m a$^{-2}$ before calving, to a higher constant rate of acceleration of 500 m a$^{-2}$ after calving, rather than a gradual increase in rate. A second step-change in acceleration occurred at the end of June 2023 following the loss of a smaller 0.25 km$^2$ iceberg from the south side of the McDonald Ice Rumples (Fig. 1b; Fig. 2g-h). Since this latest calving, GPS data from Halley VI Research Station (ZMET) show velocities at spring tide reaching 1500 m a$^{-1}$ in August 2023, almost double the maximum velocities observed over the previous sixty years (Gudmundsson, 2017). Velocities derived from speckle/feature-tracking between pairs of Sentinel-1 SAR images (Fig. 3) confirm the large increases in velocity following these calving events and show a particularly large increase in the south and southwest close to the grounding line.

## 3.2 High Strain Rate in the South

Following the calving of A-81 in January, a new area of high strain rate has developed in the south of the ice shelf near the grounding line (Fig. 3e-f). The Brunt Ice Shelf is composed of icebergs which calve off the continent via a series of small outlet glaciers and are then cemented together with sea ice and snow to form the ice shelf (King et al., 2018). The overall thickness of the incorporated icebergs is between 150 and 200 m, while the interstitial areas are metres to tens of metres thick. High velocities create large spaces between the icebergs, increasing the proportion of sea ice in the shelf and reducing the overall thickness. These thin regions have a lower resistance to tensile stress and are likely to make the ice shelf more vulnerable to the external forcings that can trigger multi-year sea ice break-up, such as swell, waves and seasonal warming (Ochwat et al., 2023). Regions of elevated strain rates have previously been observed close to the continental grounding line (DeRydt et al., 2018) with evidence of gradual consolidation over time due to snow accumulation (King et al., 2018). Since the calving of A-81, the extent and rate of opening of these areas have increased substantially and a new rift-type feature has begun propagating to the east (Fig. 3e-f)

## 3.3 Buttressing at the McDonald Bank

The long-term viability of the Brunt Ice Shelf is linked to the presence of topographic highs on the McDonald Bank which form pinning points that buttress the ice shelf and restrict its velocity. Studies of the submarine geomorphology of the McDonald Bank show a series of former pinning points have been progressively abandoned as the ice shelf has thinned and retreated towards the continent (Hodgson et al., 2019). Ice flow diversion around these pinning points has displaced high volumes of deformable sediment downstream forming wedge shaped ramps at different depths, reflecting historical changes in ice shelf draft (Arndt et al., 2019). The calving of A-81 has reduced the total area of contact between incorporated iceberg keels at the base of the Brunt Ice Shelf, the topographic high at the McDonald Ice Rumples and the deformable sediment



wedges downstream (the location of which can be inferred from observations of grounded icebergs such as the three seen in Figure 2f-h).

## 4 Discussion

The rapid acceleration of the Brunt Ice Shelf following calving represents a departure from the typical behaviour observed for this ice shelf over the last sixty years. Velocities of this magnitude are usually only measured on the ice shelves fed by exceptionally fast flowing outlet glaciers such as Thwaites, Pine Island and Denman Glaciers (Miles et al., 2020; Thompson et al., 2023). The Brunt Ice Shelf does not have a fast-flowing outlet glacier feeding it and under high velocities it is not fully replenished by ice from the continental interior. Acceleration results in thinning of the ice shelf immediately downstream of the grounding line, and we observe new areas of concentrated high strain rates in this area making the remaining ice shelf more vulnerable to external forcing. The future stability of the ice shelf now depends on when, and if, the iceberg keels incorporated within the ice shelf are able to re-establish contact with the bed at the McDonald Bank.

## 5 Conclusions

Here we present the latest data on flow and dynamics of the Brunt Ice Shelf, showing a substantial response in velocity to a large tabular iceberg calving event. The Brunt Ice Shelf is unusual in its composition, but the observations presented show that large calving events which disrupt buttressing at key pinning points can substantially and immediately impact ice flow. This has implications for other ice shelves where fractures interact with grounded margins or pinning points (e.g., Benn et al., 2022).

## 6 Data Availability

GPS data used in the manuscript are available through the UK Polar Data Centre (doi TBC). Sentinel-1 data are available through the ESA Science Hub.

## 7 Author Contributions

OJM and AJL performed the analysis of GPS and Sentinel-1 data respectively. DAH conceived the study. OJM prepared the manuscript with contributions from all authors.



**8 Competing Interests**

The authors declare that they have no conflict of interest.

**9 Acknowledgements**

We thank Dana Floricioiu and DLR for providing TerraSAR-X imagery under project HYD2997. Sentinel-1 data were
provided by the Copernicus Program of the European Commission. We thank British Antarctic Survey engineers, field guides
and glaciologists for maintaining GPS instruments on the Brunt Ice Shelf.

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



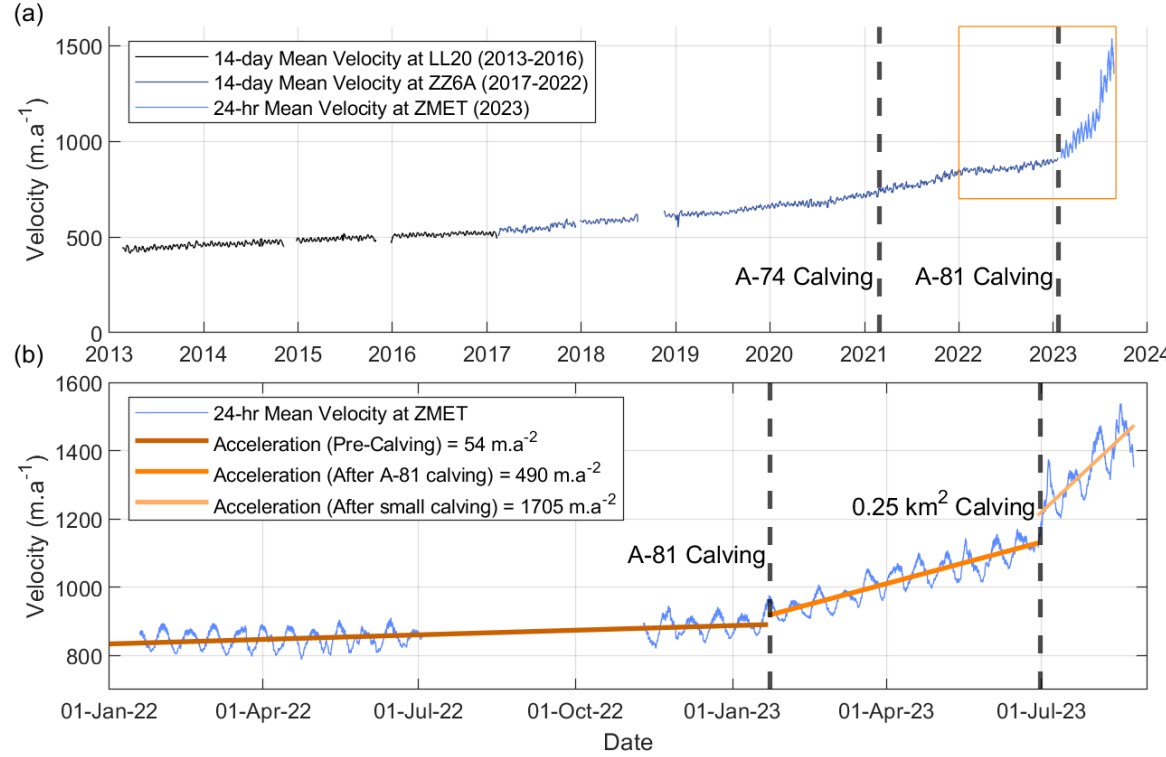

**Figure 1: a) Long-term velocities on the Brunt Ice Shelf from three GPS sites close to the current Halley VI Research Station (Fig. 2c) showing the timing of calving of A-74 and A-81 and b) step changes in acceleration rate during 2022-23 associated with calving of A-81 and a subsequent small calving event.**





**Figure 2: a) Landsat-8 from 30th December 2017 showing key features on the Brunt Ice Shelf and b) Landsat-9 from 31st March 2023 showing the locations of panels c-h, c) GPS instrument positions d-h) TerraSAR-X backscatter imagery of the McDonald Ice Rumples from d) before acceleration in 2009, e) before calving of A-81, f) after calving of A-81, g) before calving of a second small section of ice, h) after calving.**





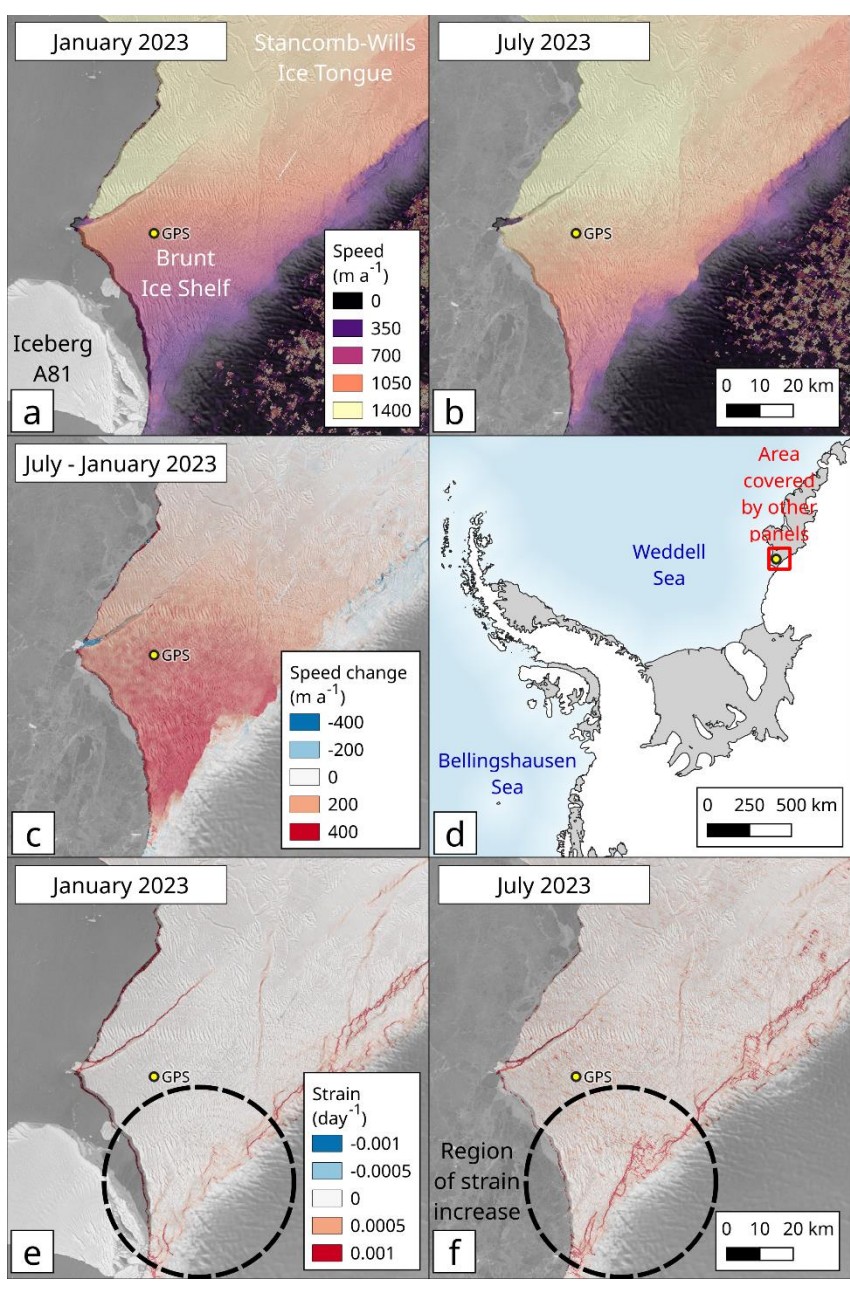

**Figure 3: (a) and (b) Velocity of the ice shelf from Sentinel-1 feature tracking in January (between images acquired on 2023-01-31 and 2023-02-12) and July (between images acquired on 2023-07-06 and 2023-07-18); (c) Observed speed up between the calving of A-81 in January 2023 and July 2023; (d) Overview map showing relevant features and area covered by other panels; (e) and (f) Magnitude of first principal strain rate derived from velocities in panels a and b showing the development of a region of high strain at the grounding line.**