# Peer review of "Brief Communication: Rapid acceleration of the Brunt Ice Shelf after calving of iceberg A-81"

_EGUsphere, 2023_

## Referee Comment (RC1)

**Brief Communication: Rapid acceleration of the Brunt Ice Shelf after calving of iceberg A-81**

*Reviewed by Chad A. Greene of NASA/JPL, Oct 10, 2023*

In this Brief Communication, Marsh et al. report on an acceleration in ice shelf velocity that occurred after the January 2023 calving of iceberg A-81. The calving event was significant enough to make news headlines when it occurred, and in this paper, the authors present a straightforward and sound analysis of the resulting ice shelf acceleration. I believe this paper is suitable for publication as a Brief Communication in *The Cryosphere*, but I think the presentation of the conclusions could be strengthened before final publication.

**Main Concerns**

My main question after reading this paper is, "Why does the calving of iceberg A-81 matter?" The most significant finding mentioned in the abstract is "*potential consequences for the stability of the remaining ice shelf*", but it's not clear what the consequences could be, when we might see them, or how likely they are to happen.

The abstract *tells* us that something may be threatening the stability of Brunt Ice Shelf, but after reading the paper, I feel like I'm missing an intuition for what exactly the threat is and how it might unfold. The closest we get is this passage:

*"...we observe new areas of concentrated high strain rates in this area making the remaining ice shelf more vulnerable to external forcing. The future stability of the ice shelf now depends on when, and if, the iceberg keels incorporated within the ice shelf are able to re-establish contact with the bed at the McDonald Bank".*

The passage above is at the heart of the main finding of this paper, but the keels are not shown in any figure, there's no diagram of any process that could stabilize or disintegrate the ice shelf, and there's no visual depiction of how close or how far we may be from catastrophic events. The authors *tell* us that the ice shelf may be under threat, but they stop short of *showing* us what that threat looks like and how serious it is. We are missing any sort of quantitative metrics for how bad the new state of vulnerability is and what exactly that will mean for the ice shelf in the future.

The language in the passage above loosely relates high strain rates to ice-shelf weakening, but it feels like the analysis stops just a little bit short of where it needs to be to come to meaningful conclusions about the fate of Brunt Ice Shelf. As a result, the conclusion of *"potential consequences for the stability of the remaining ice shelf"* seems like it's trying to raise alarms for Brunt, yet the wording is too vague to carry any real meaning.

The calving of Iceberg A-81 is a recent event, so a Brief Communication in *The Cryosphere* is a perfect place to report on it. Assuming the authors do not wish to incorporate modeling that would expand the scope of the work to a full Research Article, **I recommend publishing this paper as a Brief Communication after drawing the main conclusions into clearer focus.**

The most straightforward path to clarifying the conclusions of the paper may be found in focusing on why Brunt Ice Shelf matters, then it may be possible to perform some back-of-the-envelope calculations to support a clear conclusion related to that topic. For example, Halley Research Station is mentioned in the manuscript, but that narrative thread is never really followed. Is Halley in danger? Do the authors have any recommendations related to Halley? A Brief Communication would be a perfect place to issue guidance on this matter.

Whether it's the fate of Halley or something else that's chosen to motivate the "plot" of this manuscript, I think it will really help to shape the analysis and conclusions around something that will allow for a clear take-home message.

**Minor Comments**

**L8:** "the rate of acceleration increased by a factor of ten" It's not immediately clear what is meant by this statement. On first read, it would be easy to assume the ice was flowing at a constant velocity until the calving of A-81 caused the velocity to increase by a factor of ten. However, this is about *acceleration*, not *velocity*.

If the ice shelf was previously near steady state (constant velocity) then the acceleration would previously be near zero. This puts a near-zero value in the denominator, which makes it easy to get big numbers like a tenfold increase in acceleration. From a glaciological standpoint, I do not know that a change in acceleration has any particular meaning, so if it is in fact meaningful for some reason, that should be stated more clearly.

If I'm reading the time series of Fig 1b correctly, it looks like the velocity in Jan 2023 increased from around 900 m/yr to maybe 930 m/yr, so a ~3% increase in velocity in response to the calving of A-81? If the velocity only increased by a few percent, then the statement that "acceleration increased by a factor of ten" feels inflated for effect, unless there's some greater meaning to this statement that isn't mentioned in the abstract. Please consider rewording to clarify the main findings.

**L11:** "with potential consequences". Can we bring this statement into clearer focus? Right now, the sentence implies there's a looming threat that something major might be about to happen, but the threat is somewhat vague without a clear description of exactly what might happen, when, or why it's important. One solution may be to split the sentence in two and explicitly stating the main conclusion in the new, second sentence. Something along the lines of "...loss of

contact with the sea floor and has led to high strain rates to the south. We posit that if X happens, Y will happen." Or however the authors wish to tackle this one.

**L23-24:** The text states, "*Modelling and observations show that ice shelf thinning around Antarctica drives acceleration (Reese et al., 2018), but there are fewer observations of how tabular calving impacts ice shelves because large calving events are rare.*"

This sentence may cause a reaction among readers who will naturally think of recent calving events and all the papers that have been written about them. I recommend rewording to make it more of an "and" sentence than a "but" sentence. Something like "previous work has been done (citations), *and* we will build on it…" I don't think it's necessary to mention thinning at all, although there's no particular harm in keeping it.

**L103:** The final concluding sentence of the paper reads, "*This has implications for other ice shelves where fractures interact with grounded margins or pinning points.*"

I'm wanting a stronger concluding statement, both from a stylistic and scientific point of view. Readers are left wondering what's implied in the phrase "has implications for", I recommend rewording to state major conclusions directly and unambiguously.

**Figure order:** Figure 2 is referenced on line 20, which is before Figure 1 is referenced in the manuscript. It makes sense to reference (the current) Fig 2 first, because it's nice to establish the geographic setting before digging into the details of the time series. I recommend placing the map in Figure 1 and adjusting references to the figure numbers accordingly.

**Context map:** Right now, the spatial context map does not appear until Fig 3d, but the purpose of the context map is to help orient readers to the location of the study, so it will be most effective in the first figure that shows a map.

**Figure 1:** The caption mentions Halley Research Station, says it's shown in Fig 2c, but Halley is not labeled in Fig 2c. I recommend either labelling Halley in the map, or not mentioning it in the caption of Fig 1.

**Figure 2:** The panels of Fig 2 require some active effort on the part of the viewer to piece together what story is being told by all the different parts of the figure. The task is difficult because the eight panels mix and match three different zoom levels of overlapping regions, and it's not immediately clear how the GPS data relates to the other panels in the figure.

Aside from the panel depicting the GPS positions, the figure presents a time series of images, but panels a-b are presented out of order relative to the time series presented in panels d-h, and although the dates of panels d-h are clearly labeled, readers must go searching in the figure caption to find the dates of panels a-b. Interpreting time series is also unintuitive because the march of time zig-zags left and right from panel d through h.

I recommend finding a way to present the panels of Figure 2 in a more linear fashion. One solution may be to combine Figures 1 and 2, to present a time series of images in a line, and tie them directly to events in the velocity timeline. Below is an example from my own paper where I did something similar (https://doi.org/10.5194/tc-12-2869-2018). This example is just one idea, but the authors should feel free to take any approach that might help readers interpret the time-lapse satellite images and link them to the velocity data.

---

## Author Response (AR1)

Authors use a combination of in situ GNNS and remote sensing data to document the acceleration of the Brunt Ice Shelf (BIS) in East Antarctica, following a series of recent calving events. They postulate that observed changes in flow dynamics are linked to calving-induced loss of buttressing imposed by the McDonald Ice Rumples.

The paper is a concise and well-presented observational record of recent changes of the BIS, which are worth bringing to the attention of the glaciology community. In particular, the clear causal relationship between changes in ice-shelf geometry and the significant speed-up of the ice shelf provides an enticing natural experiment to benchmark our understanding of the role of pinning points for ice-shelf buttressing. I would like to take this opportunity to raise a few points that the authors might wish to address in a revised version of their manuscript. These are minor points, however, and overall I think the manuscript can be published in its present form.

We thank reviewer 2 for their comments.  In response to their minor points:

Comments:

Line 11: integrity instead of stability?

Line 11: Yes, we have changed this to 'strength'.

Line 48: can you reiterate here that very little of this acceleration is due to the spatial variability in the flow (GPS stations as Lagrangian trackers). Maybe refer to Gudmundsson et al. 2016.

Line 48: Yes, we have added a sentence to this effect.

Line 78: the loading mode of these rifts (shear, opening, tearing) is not unambiguously determined by the magnitude of the first principal strain rate shown in Figure 3e/f. Is it safe to assume these rifts near the grounding line are predominantly mode I (opening)?

Line 78: No, we cannot and don't assume that these are mode I type features – there is a substantial shear component across the features and they are likely a mixed mode.    The strain rate presented in Fig. 3e/f is the first principal strain as this is the best way to represent the extent of weaker areas experiencing large amounts of spreading.   There isn't additional value in presenting the second principal strains from the satellite data, which don't influence our interpretation.

Fig 1b. Can you highlight the time period over which the satellite date in Fig3a-c were collected?

Fig 1b: Yes, we have marked the images used in Fig. 3 onto Fig. 1 (now Figure 2).

Fig 2d-h. From the velocity data or other proxies, do you have an estimate of where the McDonald Ice Rumples are grounded, and can you draw a grounding line on these figures for some further context?

Fig 2d-h: The shear associated with grounding around the McDonald Ice Rumples creates crevassing which makes it not possible to calculate an interferometric grounding line. We have marked an approximate zero-velocity contour - this is not likely to match exactly to the grounding line, but is the best we can do here.

Fig 3c. The observed spatial distribution of speed changes looks very different from numerical experiments that were designed to test the flow response to ungrounding of the McDonald Ice Rumples. E.g. Gudmundsson et al. (2016) in their figure 10 show that numerical simulations project the largest speed-up occurs near the Ice Rumples. Presumably (part of) this disagreement can be related to the lack of an evolving ice damage field in the model. Can you comment on the disagreement between existing numerical simulations and observations, and can you suggest possible approaches to reduce that mismatch? Maybe this could be an interesting preamble to 'future work'.

Fig 3c: Yes, we have written a short comment on differences between observations and previous modelling efforts. I think the discrepancy is in large part due to difficulty in inverting for the highly variable viscosity field caused by 'blocks' and 'infill' close to the grounding line. This variability in ice properties is described by King et al., 2018, and we have highlighted this complexity in the discussion.

In this Brief Communication, Marsh et al. report on an acceleration in ice shelf velocity that occurred after the January 2023 calving of iceberg A-81. The calving event was significant enough to make news headlines when it occurred, and in this paper, the authors present a straightforward and sound analysis of the resulting ice shelf acceleration. I believe this paper is suitable for publication as a Brief Communication in The Cryosphere, but I think the presentation of the conclusions could be strengthened before final publication.

Main Concerns My main question after reading this paper is, "Why does the calving of iceberg A-81 matter?" The most significant finding mentioned in the abstract is "potential consequences for the stability of the remaining ice shelf", but it's not clear what the consequences could be, when we might see them, or how likely they are to happen. The abstract tells us that something may be threatening the stability of Brunt Ice Shelf, but after reading the paper, I feel like I'm missing an intuition for what exactly the threat is and how it might unfold. The closest we get is this passage: "…we observe new areas of concentrated high strain rates in this area making the remaining ice shelf more vulnerable to external forcing. The future stability of the ice shelf now depends on when, and if, the iceberg keels incorporated within the ice shelf are able to re-establish contact with the bed at the McDonald Bank". The passage above is at the heart of the main finding of this paper, but the keels are not shown in any figure, there's no diagram of any process that could stabilize or disintegrate the ice shelf, and there's no visual depiction of how close or how far we may be from catastrophic events. The authors tell us that the ice shelf may be under threat, but they stop short of showing us what that threat looks like and how serious it is. We are missing any sort of quantitative metrics for how bad the new state of vulnerability is and what exactly that will mean for the ice shelf in the future. The language in the passage above loosely relates high strain rates to ice-shelf weakening, but it feels like the analysis stops just a little bit short of where it needs to be to come to meaningful conclusions about the fate of Brunt Ice Shelf. As a result, the conclusion of "potential

consequences for the stability of the remaining ice shelf" seems like it's trying to raise alarms for Brunt, yet the wording is too vague to carry any real meaning.

We thank reviewer 1 for their comments. It is a valid question to ask why this particular calving matters, but it is not just the calving itself but the quality of data around the calving and mechanism that matters. The data presented is relatively rare in capturing the precise timing of calving (to the hour), of which I can only think of one other example (Banwell et al., 2017), and capturing high-resolution of the immediate response of the remaining ice shelf, of which previous data shows responses developing slowly in the years following calving or ice shelf collapse (e.g. Scambos et al., 2004), instead of immediately (as here).

The behaviour of this ice shelf is of additional interest to the UK's Antarctic research community, given the location of the Halley VI research station. The calving, leading to a doubling of velocity is an unprecedented change for this ice shelf, although we do not want to make this the primary focus of the paper and think the mechanism and response rate is of more interest to the wider community.

We unfortunately do not have good data on keel depths and are unable to reliably model the future evolution of fractures in the ice shelf at this stage (hence why this is published as a Brief Communication). From current velocities and locations of other grounding icebergs we can make some inferences on the latest possible time of re-grounding, which have added to the conclusions, but the range of scenarios for the future of the ice shelf have high uncertainty, and in our opinion, it is not useful to speculate at this stage.

The calving of Iceberg A-81 is a recent event, so a Brief Communication in The Cryosphere is a perfect place to report on it. Assuming the authors do not wish to incorporate modeling that would expand the scope of the work to a full Research Article, I recommend publishing this paper as a Brief Communication after drawing the main conclusions into clearer focus. The most straightforward path to clarifying the conclusions of the paper may be found in focusing on why Brunt Ice Shelf matters, then it may be possible to perform some back-of-theenvelope calculations to support a clear conclusion related to that topic. For example, Halley Research Station is mentioned in the manuscript, but that narrative thread is never really followed. Is Halley in danger? Do the authors have any recommendations related to Halley? A Brief Communication would be a perfect place to issue guidance on this matter. Whether it's the fate of Halley or something else that's chosen to motivate the "plot" of this manuscript, I think it will really help to shape the analysis and conclusions around something that will allow for a clear take-home message.

Advice around operation decisions for Halley VI involve many factors in addition to the glaciology which we cannot comment on here. The main message of the paper is the rapid response observed in the shelf after calving, and that this may be missed in satellite-only analyses of calving. We have strengthened this message.

Minor Comments L8: "the rate of acceleration increased by a factor of ten" It's not immediately clear what is meant by this statement. On first read, it would be easy to assume the ice was flowing at a constant velocity until the calving of A-81 caused the velocity to increase by a factor of ten. However, this is about acceleration, not velocity. If the ice shelf was previously near steady state (constant velocity) then the acceleration would previously be near zero. This puts a near-zero value in the denominator, which makes it easy to get big numbers like a tenfold increase in acceleration. From a glaciological standpoint, I do not know that a change in acceleration has any particular meaning, so if it is in fact meaningful for some reason, that should be stated more clearly. If I'm

reading the time series of Fig 1b correctly, it looks like the velocity in Jan 2023 increased from around 900 m/yr to maybe 930 m/yr, so a ~3% increase in velocity in response to the calving of A-81? If the velocity only increased by a few percent, then the statement that "acceleration increased by a factor of ten" feels inflated for effect, unless there's some greater meaning to this statement that isn't mentioned in the abstract. Please consider rewording to clarify the main findings.

We agree that there is possibility for confusion between changes in velocity over time (with specified time frames) and accelerations (rates of change of velocity without a specified time frame), which essentially have the same units. Changes in velocity over time are certainly more common in glaciology. Here though, velocities fluctuate with the fortnightly tidal variability and immediate changes in velocity over very short timescales (days) are masked by the variability, while changes over longer periods are heavily dependent on the length of the period.

We present rates of acceleration instead, which show clear changes coinciding with calving - the acceleration rate is shown by the linear fits in Fig 1b. The three different fits are punctuated by clearly observable events (two separate calving events) and the change in acceleration from one constant rate to another implies a discrete change in dynamics. There is a 'ten-fold increase' after the first event (which on its own may not be that useful) but it is useful to compare with the 3-fold increase after the second calving, to give a sense of the magnitude of the change in dynamics. We have modified the abstract to make it clearer than we are primarily discussing acceleration, not velocity.

L11: "with potential consequences". Can we bring this statement into clearer focus? Right now, the sentence implies there's a looming threat that something major might be about to happen, but the threat is somewhat vague without a clear description of exactly what might happen, when, or why it's important. One solution may be to split the sentence in two and explicitly stating the main conclusion in the new, second sentence. Something along the lines of "...loss of 3 contact with the sea floor and has led to high strain rates to the south. We posit that if X happens, Y will happen." Or however the authors wish to tackle this one.

We have removed the reference to "potential consequences" but discuss future scenarios (dependence on re-grounding) without being too speculative.

L23-24: The text states, "Modelling and observations show that ice shelf thinning around Antarctica drives acceleration (Reese et al., 2018), but there are fewer observations of how tabular calving impacts ice shelves because large calving events are rare." This sentence may cause a reaction among readers who will naturally think of recent calving events and all the papers that have been written about them. I recommend rewording to make it more of an "and" sentence than a "but" sentence. Something like "previous work has been done (citations), and we will build on it…" I don't think it's necessary to mention thinning at all, although there's no particular harm in keeping it.

There are a number of papers about recent calving events, but not about the direct impacts on the remaining ice shelves. We have removed the reference to Reese et al., 2018, and instead cite some work on the effect of loss of buttressing on ice flow.

L103: The final concluding sentence of the paper reads, "This has implications for other ice shelves where fractures interact with grounded margins or pinning points." I'm wanting a stronger concluding statement, both from a stylistic and scientific point of view. Readers are left wondering

what's implied in the phrase "has implications for", I recommend rewording to state major conclusions directly and unambiguously.

Yes, we have reworded this to emphasise the direct and rapid impact of calving close to pinning points (i.e. the implications are that a calving event can completely alter the dynamics of an ice shelf with immediate effect).

Figure order: Figure 2 is referenced on line 20, which is before Figure 1 is referenced in the manuscript. It makes sense to reference (the current) Fig 2 first, because it's nice to establish the geographic setting before digging into the details of the time series. I recommend placing the map in Figure 1 and adjusting references to the figure numbers accordingly. Context map: Right now, the spatial context map does not appear until Fig 3d, but the purpose of the context map is to help orient readers to the location of the study, so it will be most effective in the first figure that shows a map.

We have swapped the order of Figures 1 and 2 and added the context map to Figure 2 (now Fig. 1).

Figure 1: The caption mentions Halley Research Station, says it's shown in Fig 2c, but Halley is not labeled in Fig 2c. I recommend either labelling Halley in the map, or not mentioning it in the caption of Fig 1.

We have removed the mention of Halley from the caption.

Figure 2: The panels of Fig 2 require some active effort on the part of the viewer to piece together what story is being told by all the different parts of the figure. The task is difficult because the eight panels mix and match three different zoom levels of overlapping regions, and it's not immediately clear how the GPS data relates to the other panels in the figure. Aside from the panel depicting the GPS positions, the figure presents a time series of images, but panels a-b are presented out of order relative to the time series presented in panels d-h, and although the dates of panels d-h are clearly labeled, readers must go searching in the figure caption to find the dates of panels a-b. Interpreting time series is also unintuitive because the march of time zig-zags left and right from panel d through h. 4 I recommend finding a way to present the panels of Figure 2 in a more linear fashion. One solution may be to combine Figures 1 and 2, to present a time series of images in a line, and tie them directly to events in the velocity timeline. Below is an example from my own paper where I did something similar (https://doi.org/10.5194/tc-12-2869-2018). This example is just one idea, but the authors should feel free to take any approach that might help readers interpret the time-lapse satellite images and link them to the velocity data.

We have modified Figures 1 and 2 (to include a context map) and indicated the timings of panels in Figure 1 and Figure 3, on the timeline in Figure 2.  We have changed the layout of Figure 2 (now Figure 1) to 3-panels across, hopefully making the progression through the images of the McDonald Ice Rumples easier to follow.